# Identification of therapeutics that target eEF1A2 and upregulate utrophin A translation in dystrophic muscles

Christine Péladeau[1,2], Nadine Adam[1,2], Lucas M. Bronicki[1,2], Adèle Coriati[1], Mohamed Thabet[1], Hasanen Al-Rewashdy[1,2], Jason Vanstone [3], Alan Mears[3], Jean-Marc Renaud[1], Martin Holcik[4] & Bernard J. Jasmin[1,2✉]

Up-regulation of utrophin in muscles represents a promising therapeutic strategy for the treatment of Duchenne Muscular Dystrophy. We previously demonstrated that eEF1A2 associates with the 5′UTR of utrophin A to promote IRES-dependent translation. Here, we examine whether eEF1A2 directly regulates utrophin A expression and identify via an ELISA-based high-throughput screen, FDA-approved drugs that upregulate both eEF1A2 and utrophin A. Our results show that transient overexpression of eEF1A2 in mouse muscles causes an increase in IRES-mediated translation of utrophin A. Through the assessment of our screen, we reveal 7 classes of FDA-approved drugs that increase eEF1A2 and utrophin A protein levels. Treatment of mdx mice with the 2 top leads results in multiple improvements of the dystrophic phenotype. Here, we report that IRES-mediated translation of utrophin A via eEF1A2 is a critical mechanism of regulating utrophin A expression and reveal the potential of repurposed drugs for treating DMD via this pathway.

[1] Department of Cellular and Molecular Medicine, Faculty of Medicine, University of Ottawa, 451 Smyth Road, Ottawa, ON K1H 8M5, Canada. [2] Centre for Neuromuscular Disease, 451 Smyth Road, Ottawa, ON K1H 8M5, Canada. [3] Apoptosis Research Centre, Children's Hospital of Eastern Ontario Research Institute, 401 Smyth Road, Ottawa, ON K1H 5B2, Canada. [4] Department of Health Sciences, Carleton University, 1125 Colonel By Drive, Ottawa, ON K1S 5B6, Canada. ✉email: jasmin@uottawa.ca

Duchenne muscular dystrophy (DMD) is the most common hereditary debilitating muscle disease and is caused by the absence of dystrophin protein in skeletal muscle[1]. The major functional role of dystrophin is to create a link between the internal cytoskeletal actin network and the extracellular matrix in order to provide structural integrity to the sarcolemma of muscle fibers[2,3]. Skeletal muscle fibers lacking dystrophin, as observed in DMD patients, display a higher susceptibility to stress-induced sarcolemmal injury, extracellular calcium influx in muscle, increased inflammation and replacement of muscle fibers by connective and adipose tissues[4]. DMD patients eventually succumb to the disease by early adulthood due to cardiac or respiratory failure[5,6]. Despite the fact that over 200 promising studies, diagnostic tests and treatment trials are currently in various stages of enrollment (the NIH Clinical Trials.gov), there is still no effective treatment. One potential therapy aims to increase the sarcolemmal expression of utrophin A, the autosomal homolog of dystrophin, that can functionally compensate for its loss in muscles of several animal models of DMD[7,8]. Several studies have indeed demonstrated using either transgenic[7,8] or pharmacological[9–14] strategies that enhancing expression of utrophin A can alleviate numerous pathophysiological features of DMD and can thus be of great therapeutic benefit.

Over the years, multiple studies have focused on determining the key transcriptional regulatory mechanisms that control utrophin A expression in muscle[15–22]. However, recent evidence demonstrates the importance of post-transcriptional and translational events in the regulation of utrophin A. In fact, expression of utrophin A has been shown to be highly regulated at its 3' end, where cis-elements promote the stability of utrophin A mRNA transcripts[23–26]. In contrast, other studies have demonstrated discordance between utrophin A protein and mRNA levels in DMD muscle biopsy samples and mouse regenerating muscle fibers, suggesting that utrophin A expression is also regulated at the translational level[26,27]. In this context, the *utrophin* gene can produce two full-length isoforms, utrophin A and utrophin B, which are transcribed from distinct promoters and have different 5'-untranslated regions (5'-UTRs)[16,28]. Both proteins are identical, except for N-terminal regions[28]. The 5'UTR of utrophin A, the skeletal muscle isoform, is long and CG-rich which suggests that utrophin A can indeed be subjected to translational control as long CG-rich elements can reduce the efficiency of conventional scanning from the 5'-end during cap-dependent protein translation[16,29–31]. Our laboratory has discovered some years ago the presence of an internal ribosome entry site (IRES) within the 5'UTR of utrophin A that promotes expression through IRES-dependent translational mechanisms[31,32]. Of relevance, our initial findings have been confirmed by others[29] and, in addition, an IRES was found in the dystrophin transcript[33].

The rate-limiting step of cap-dependent translational initiation is the binding of the eukaryotic initiation factor (EIF) 4F protein complex to the 7-methylguanylate cap ($m^7G$), also known as the 5'cap. Under certain cellular and physiological conditions, including disease or stress, IRES-dependent translation of mRNAs is enhanced while cap-dependent translation is simultaneously attenuated[34,35]. IRES elements are thought to associate with the translational machinery, including the canonical initiation factors, as well as IRES trans-acting factors (ITAFs), which enable the recruitment of the ribosome to initiate peptide synthesis[30,36]. It has been suggested that ITAFs act as RNA chaperones to modulate IRES activity in the appropriate conformational formation to promote ribosome binding[37]. However, the precise mechanisms involved in IRES-dependent translation remain largely unknown.

Our laboratory previously demonstrated that muscles expressing a bicistronic reporter construct containing the utrophin A 5'

UTR and subjected to degeneration and regeneration cycles by cardiotoxin injections, generated strong utrophin A IRES activity[31]. In addition to potential translational events regulating utrophin A in regenerating fibers, our laboratory also demonstrated activation of this IRES following glucocorticoid treatment[12,31]. Interestingly, this IRES appears capable of preferentially driving the translation of utrophin A in skeletal muscle[32]. Through a series of experiments including RNA-affinity chromatography, mass spectrometry and UV-crosslinking studies, we previously identified eEF1A2 as a putative ITAF able to modulate the activity of the utrophin A IRES[32]. Our aims in the present study are three-fold. First, we wish to examine the role of eEF1A2 in directly regulating the endogenous expression of utrophin A in muscle of several mouse models. Next, by performing a high-throughput drug screen, we sought to identify FDA-approved drugs that target eEF1A2, thereby upregulating utrophin A expression through IRES activation. Finally, we want to characterize the therapeutic potential of activating the translation of utrophin A through eEF1A2 in mdx mouse muscle with leads identified in the screen. Collectively, we identify several FDA-approved drugs that stimulate IRES-dependent translation of utrophin A through eEF1A2, with potential to accelerate the clinical implementation of therapeutics to treat DMD. Our findings provide several complementary physiological lines of evidence indicating that targeting the activity of the utrophin A IRES is a viable strategy with potential therapeutic benefits for increasing endogenous expression of utrophin A in DMD muscle fibers.

## Results

**Expression of eEF1A2 in fast and slow muscles of mdx mice.** In a first set of experiments, we examined whether the endogenous expression of eEF1A2 differs in wild-type versus mdx (a DMD mouse model) mice, in fast extensor digitorum longus (EDL) and slow soleus muscles. Mdx and wild-type mouse muscle lysates were used for western blot analyses. Results did not reveal any significant ($P > 0.05$, two-tailed Student's $t$-test) difference in the relative abundance of eEF1A2 protein content in fast and slow muscles of wild-type versus mdx mice (Fig. 1a). This indicates that strategies aimed at further increasing the expression and/or activity of eEF1A2 in muscle may be of therapeutic benefit for DMD patients.

**Overexpression of eEF1A2 in muscle increases utrophin A.** To determine the impact of eEF1A2 on utrophin A expression in vivo, we overexpressed eEF1A2 in skeletal muscle and analyzed utrophin A protein levels in both wild-type and mdx mice. To this end, we electroporated an eEF1A2-expressing construct (MYC-HIS360-eEF1A2-pcDNA) or pcDNA3.1 control into the tibialis anterior (TA) muscles of wild-type and mdx mice and harvested the muscles 7 days later. Western blot analyses showed a nearly 2-fold increase ($P < 0.05$, two-tailed Student's $t$-test) in utrophin A protein levels in wild-type and mdx mouse muscles overexpressing eEF1A2 as compared to controls (Fig. 1b). This result is exciting because only a fraction (~20%) of the muscle fibers expresses the eEF1A2 expression plasmid under these injection/electroporation conditions.

In these experiments, it was important to determine whether the increase in utrophin A protein levels seen following eEF1A2 overexpression is mediated via activation of the utrophin A IRES. Accordingly, we electroporated the eEF1A2-expressing construct into TA muscles of transgenic mice previously generated in our lab[32], which harbor the CMV/β-GAL/UtrA/CAT bicistronic reporter transgene that contains the utrophin A 5'UTR. In this construct, the first cistron of β-galactosidase (β-GAL) reflects

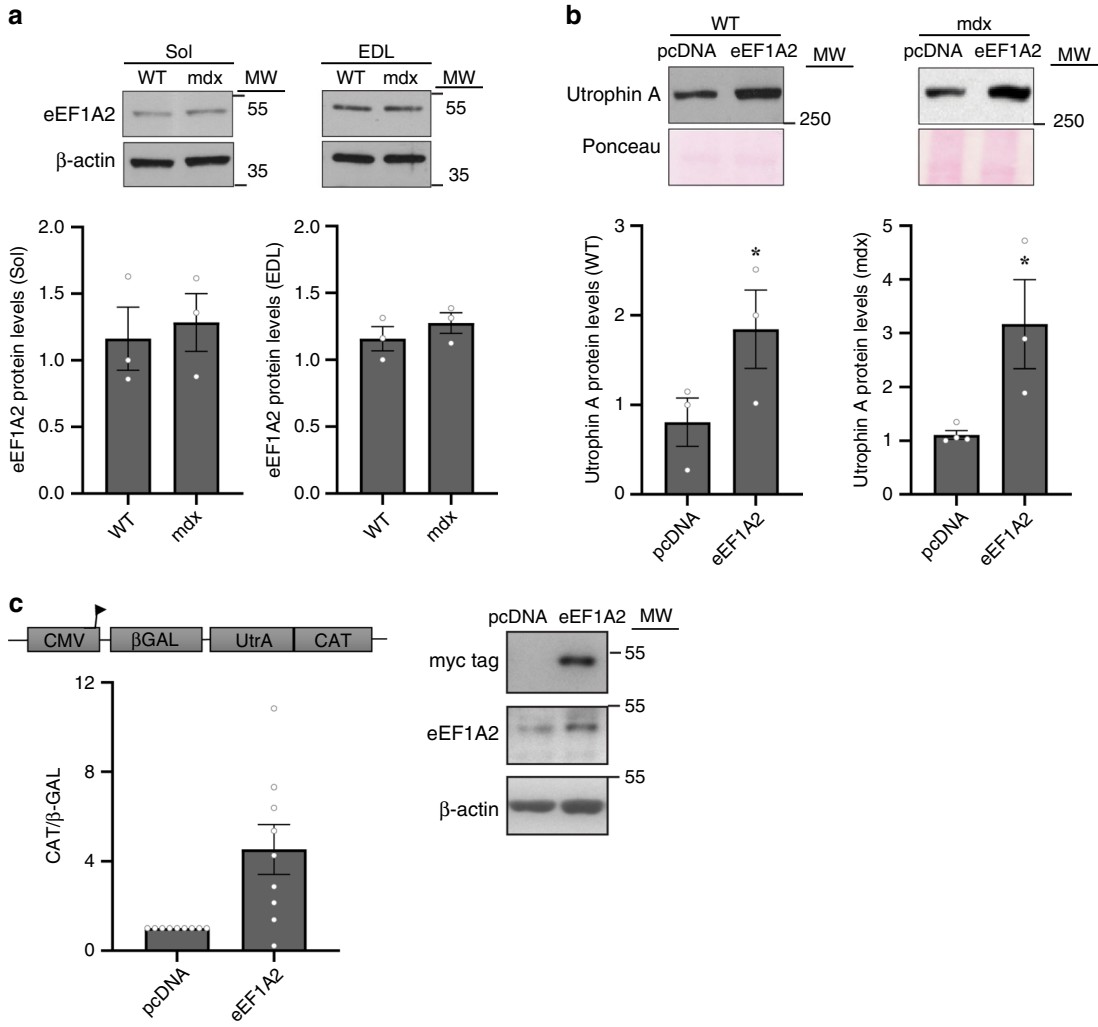

**Fig. 1 Overexpression of eEF1A2 in skeletal muscles increases endogenous utrophin A protein levels and utrophin A IRES activity. a** Representative western blot comparing the expression profile of eEF1A2 protein in fast (EDL) and slow (soleus) skeletal muscles harvested from 6 to 7-week-old wild-type (WT) and mdx mice. β-actin was used as a loading control ($N = 3$). **b** eEF1A2 (eEF1A2-pcDNA) expression construct or control (pcDNA3.1) were electroporated into TA muscles of wild-type and mdx mice as well as in TA muscles from utrophin A 5′UTR (CMV/βGAL/UtrA/CAT) reporter transgenic mice. Representative western blots of endogenous utrophin A protein expression in wild-type and mdx mice following overexpression of eEF1A2 with the respective quantification. Ponceau staining was used as a loading control. Note the increase in utrophin A protein levels in skeletal muscles overexpressing eEF1A2 ($N = 3$). **c** Relative IRES activity as determined by a ratio of CAT: β-GAL activity in TA muscles of utrophin A 5′UTR (CMV/βGAL/UtrA/CAT) transgenic mice overexpressing eEF1A2 ($N = 9$). On the right, representative western blots demonstrating expression levels of the myc tag containing pcDNA-eEF1A2 expression vector and eEF1A2 expression levels in TA transgenic mouse muscle. Error bars represent SEM. *$P < 0.05$, versus control. Two-tailed Student's $t$-test were performed to determine statistical differences in these experiments. Source data are provided as a Source Data file.

cap-dependent translation whereas the second cistron chloramphenicol acetyltransferase (CAT) represents IRES-dependent translation regulated by the inserted utrophin A 5′UTR. Therefore, an increase in CAT activity accompanied by no changes in β-GAL corresponds to an activation of utrophin A IRES-dependent translation. Following electroporation of the eEF1A2 construct into TA muscles, we performed a series of standard analyses to determine utrophin A IRES activity as shown by a ratio of CAT to β-GAL activity. Our analysis demonstrated a clear trend toward increased IRES reporter activity in muscles of our transgenic mice overexpressing eEF1A2 (Fig. 1c). Due to the variability inherent to these experiments, this change did not reach statistical significance ($P = 0.19$, two-tailed Student's $t$-test). However, examination of the averaged raw values of reporter activity in muscles overexpressing eEF1A2 and control showed, as expected for IRES activation, an increase in CAT (control;

$161.3 \pm 31.2$ versus eEF1A2; $302.3 \pm 33.3$, $P = 0.17$, two-tailed Student's $t$-test), but little change in β-GAL (control $0.55 \pm 0.12$ versus eEF1A2; $0.61 \pm 0.14$, $P = 0.76$, two-tailed Student's $t$-test) activity. Collectively, these findings indicate that eEF1A2 directly increases endogenous utrophin A protein expression by acting via the utrophin A IRES.

**Pharmacological activation of utrophin A through eEF1A2.** Based on our data demonstrating that overexpression of eEF1A2 stimulates utrophin A protein expression in skeletal muscle of wild-type and mdx mice, we sought to identify FDA-approved drugs that target eEF1A2 to activate utrophin A translation as a therapeutic approach for treating DMD. Accordingly, we designed an ELISA-based high-throughput drug screen with a total of 262 FDA-approved drugs. C2C12 myoblasts were treated with each drug or vehicle control for 24 h. It is

important to note that the drug doses used in this screen are clinically relevant. Following treatment, protein levels of both eEF1A2 and utrophin A were assessed. A drug was considered a hit if it had the ability to significantly raise both eEF1A2 and utrophin A protein levels over vehicle control. From this screen, we obtained 11 drugs that were considered as leads (Table 1 and Fig. 2a).

We noted that some of these FDA-approved drugs have common roles such as anti-diabetic, anti-peptic ulcer, cholesterol-lowering and beta-adrenergic blocking agents (Table 1). Interestigly, we found in this high-throughput screen that four drugs, Acarbose, Labetalol, Pravastatin, and Telbivudine (Table 1 and Fig. 2a) caused significant and reproducible increases of at least ~2-fold in the levels of eEF1A2 and utrophin A. This is important because it has been shown that a 2-fold increase of utrophin A in muscle is sufficient to improve the dystrophic pathology[7]. Subsequent confirmation of the effects of the 11 drugs was performed by treating C2C12 myoblasts with each drug at three different doses, including the dose used for the screen, for a period of 24 h. Western blot analysis showed that under these conditions, expression of both utrophin A and eEF1A2 increased significantly ($P < 0.05$, one-way ANOVA) to an extent similar to or higher than that observed in the high-throughput screen, thereby validating our findings obtained with the screen (Table 2

**Table 1 eEF1A2 and utrophin A activating FDA-approved drugs identified using an ELISA-based high-throughput screen.**

| Compound name | Compound family |
|---|---|
| Rosiglitazone | Anti-diabetic agent |
| Acarbose | |
| Nizatidine | Anti-peptic ulcer agent |
| Olsalazine·Na | |
| Lovastatin | Cholesterol-lowering agent |
| Pravastatin·Na | |
| Betaxolol·HCl | Beta-adrenergic blocking agent |
| Labetalol | |
| Hydrochlorothiazide | Diuretic agent |
| Propylthiouracil | Anti-hyperthyroidism agent |
| Telbivudine | Anti-hepatitis B virus agent |

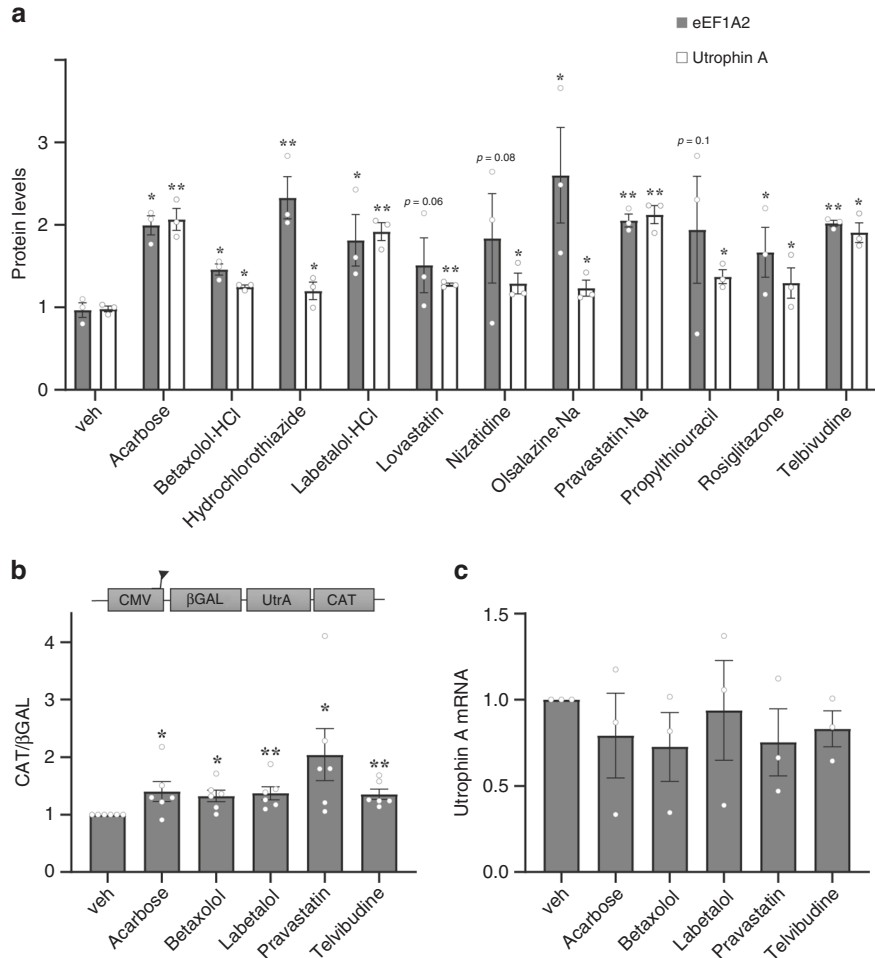

**Fig. 2 Pharmacological activation of eEF1A2 and utrophin A by FDA-approved drugs. a** An ELISA-based high-throughput drug screen was performed by treating C2C12 cells with 262 different FDA-approved drugs or vehicle control for 24 hours. Quantification of eEF1A2 and utrophin A protein levels from the 11 FDA-approved drugs considered as leads. A drug is considered a hit based on its ability to increase eEF1A2 and utrophin A protein levels over vehicle control ($N = 3$). **b** Activation of the utrophin 5′UTR IRES reporter construct by 24-h treatment of FDA-approved drugs in C2C12 cells. The treated muscle cell samples were subjected to a reporter assay to determine CAT and β-GAL activity representative of IRES activity. CAT activation was normalized to β-GAL and each drug activation was normalized to vehicle control ($N = 6$). **c** Relative quantification of utrophin A mRNA levels in C2C12 treated cells, determined by qRT-PCR ($N = 3$). The values were normalized to 18S mRNA levels. Error bars represent SEM. \*$P < 0.05$, \*\*$P < 0.01$, versus vehicle control. One-way ANOVA and Bonferroni post-hoc test was performed to determine statistical differences. Source data are provided as a Source Data file.

**Table 2 Confirmation of eEF1A2 and utrophin A protein expression level increases in C2C12 cells, post FDA-approved drug treatment.**

| Drug | Utrophin A protein level (fold increase to veh ctl) | eEF1A2 protein level (fold increase to veh ctl) | Optimal concentration |
|---|---|---|---|
| Acarbose | 2.51 | 4.62 | 1 µM |
| Betaxolol | 2.13 | 4.26 | 5 µM |
| Labetalol·HCl | 3.25 | 1.41 | 1 µM |
| Telbivudine | 2.72 | 2.77 | 30 µM |
| Pravastatin·Na | 1.84 | 2.30 | 200 nM |
| Olsalazine·Na | 1.58 | 1.13 | 4 µM |
| Lovastatin | 1.34 | 1.12 | 100 nM |
| Nizatidine | 1.22 | No increase | 1 µM |
| Propylthiouracil | 1.17 | No increase | 10 µM |
| Hydrochlorothiazide | No increase | 1.44 | 5 µM |
| Rosiglitazone | No increase | 1.30 | 5 µM |

The drugs above the middle horizontal line are the five top drugs picked for further analysis in vitro and in vivo.

and Supplemental Fig. 1). With the high sensitivity and quantitative analysis of the ELISA assay used in the high-throughput screen, we were able to easily detect significant increases of eEF1A2 and utrophin A at lower doses than the western blot analyses. However, despite using two distinct technical approaches and despite the limitations of western blot analyses, we observed similar results to the ELISA and determined that most drugs were inducing a dose-dependent response of eEF1A2 and utrophin A protein levels. This indicates that our screen to identify drugs targeting eEF1A2 and utrophin A proved to be successful at finding FDA-approved drugs that could be beneficial for the treatment of DMD.

**Pharmacological stimulation of utrophin A IRES in vitro.** We wanted to verify whether the potent eEF1A2-targeting drugs considered as leads in our screen acted by stimulating IRES-mediated translation of utrophin A. For these experiments, we focused on the five leads that maximally upregulate eEF1A2 and utrophin A protein levels (Acarbose, Betaxolol, Labetalol, Pravastatin and Telbivudine), as determined by western blots (Table 2). Thus, we transfected cultured C2C12 myoblasts with a bicistronic construct, either the control CMV/βGAL/CAT or the CMV/βGAL/UtrA/CAT containing the utrophin A 5′UTR, and treated cells for 24 h with vehicle control or one of the five top utrophin A- and eEF1A2-activating drugs. Reporter activity from cell lysates was assessed by using a CAT and β-GAL ELISA reporter assay kit. The activity of the utrophin A IRES was determined by establishing a ratio of CAT to β-gal activity. Our data showed a significant ~1.35 to 2-fold increase in CAT/β-gal ratios ($P \leq 0.05$ and $P \leq 0.01$, one-way ANOVA) in which CAT levels increased and β-GAL levels remained constant, thus demonstrating that the five top FDA-approved drugs did, in fact, activate utrophin A through IRES-mediated translation (Fig. 2b). In addition, none of these drugs increased utrophin A mRNA levels, further suggesting activation of utrophin A through translational events (Fig. 2c).

**Pharmacological activation of utrophin A IRES in vivo.** In a next series of experiments, we examined whether the five top drugs identified in the high-throughput screen cause upregulation of utrophin A via eEF1A2 thereby promoting IRES-mediated translation of utrophin A in vivo. To this end, we treated our

IRES-transgenic mice (CMV/βGAL/UtrA/CAT) with one of the five drugs or a vehicle control for a period of 7 days. After the treatment period, we observed that Pravastatin (2 mg/kg) and Betaxolol (5 mg/kg) increased endogenous levels of eEF1A2 (~2.5-fold) and utrophin A (~2 to 2.5-fold) in TA muscles (Fig. 3a, b). Note that these doses were selected in reference to past preclinical studies in mice and their clinical use in humans[38,39]. Western blot analyses of CAT and β-GAL were used to establish a ratio of CAT vs β-GAL activity to determine utrophin A IRES activity following treatment with Betaxolol or Pravastatin of transgenic mice. Despite some variability, Betaxolol and Pravastatin treatment resulted in a clear trend toward an increase in IRES reporter activity in muscles from these transgenic mice (~1.5 to 2-fold; $P = 0.3$ and $P = 0.6$, one-way ANOVA, respectively, Fig. 3c). Furthermore, utrophin mRNA levels were unchanged post-treatment (Fig. 3d), thus indicating that both drugs increased endogenous utrophin A protein levels in vivo by acting via the utrophin A IRES.

**Betaxolol and Pravastatin enhance strength of mdx mice.** To establish whether Betaxolol and Pravastatin have therapeutic benefits in vivo, we treated 6-week old mdx and wild-type mice with either Pravastatin (2 mg/kg), Betaxolol (5 mg/kg) or saline for 4 weeks and analyzed the performance of the mice at the end of the treatment period. For this, mice were subjected to a digital gauge attached to a grid to detect forelimb and hindlimb grip strength. The weight of the mice was assessed throughout the 4-week treatment period and did not show any difference between drug- and vehicle-treated mdx mice (Fig. 4a). Importantly, mdx mice treated for 4 weeks with Betaxolol showed significant improvements in forelimb grip strength ($P < 0.05$, one-way ANOVA) as well as a trend toward an increase in hindlimb grip strength ($P = 0.08$, one-way ANOVA) when normalizing for mouse weight (Fig. 4B,C). The 4-week Pravastatin treatment of mdx mice induced striking ameliorations in forelimb grip strength ($P < 0.01$, one-way ANOVA), rescuing it near wild-type levels, together with a significant increase in hindlimb grip strength ($P < 0.01$, one-way ANOVA) with or without adjusting for mouse weight (Fig. 4b, c). Overall, this demonstrates that both eEF1A2-targeting drugs improve muscle strength of dystrophic mice.

After 4–6 weeks of treatment, wild-type as well as Betaxolol-, Pravastatin- and saline-treated mdx mice were euthanized. With these mice, we performed ex vivo experiments in which an EDL muscle from mice of each group was subjected to a series of eccentric contractions. Our results show that treatment of mdx mice with Betaxolol and Pravastatin significantly ($P < 0.05$, split-plot ANOVA) improved force-drop during the 12 eccentric contractions (Fig. 4d). These data along with our in vivo force measurements above, demonstrate that both drugs clearly improve dystrophic muscle function.

**Betaxolol and Pravastatin increase sarcolemmal utrophin A.** After the 4-week treatment with Betaxolol, Pravastatin or vehicle control, TA muscles from wild-type and mdx-treated mice were dissected and cross-sectioned for further analyses. Since it is essential for utrophin A to localize to the sarcolemma in order to fully achieve its function[40], we examined utrophin A's sarcolemmal localization in vehicle-, Betaxolol- and Pravastatin-treated TA muscles. As expected, utrophin A is localized at the neuromuscular junction in wild-type muscles[41,42]. Immunofluorescence analysis and quantification of positive fibers showing sarcolemmal utrophin A demonstrated that both Betaxolol and Pravastatin treatments nearly doubled the expression of utrophin A at the sarcolemma compared to vehicle control (Fig. 5a, b).

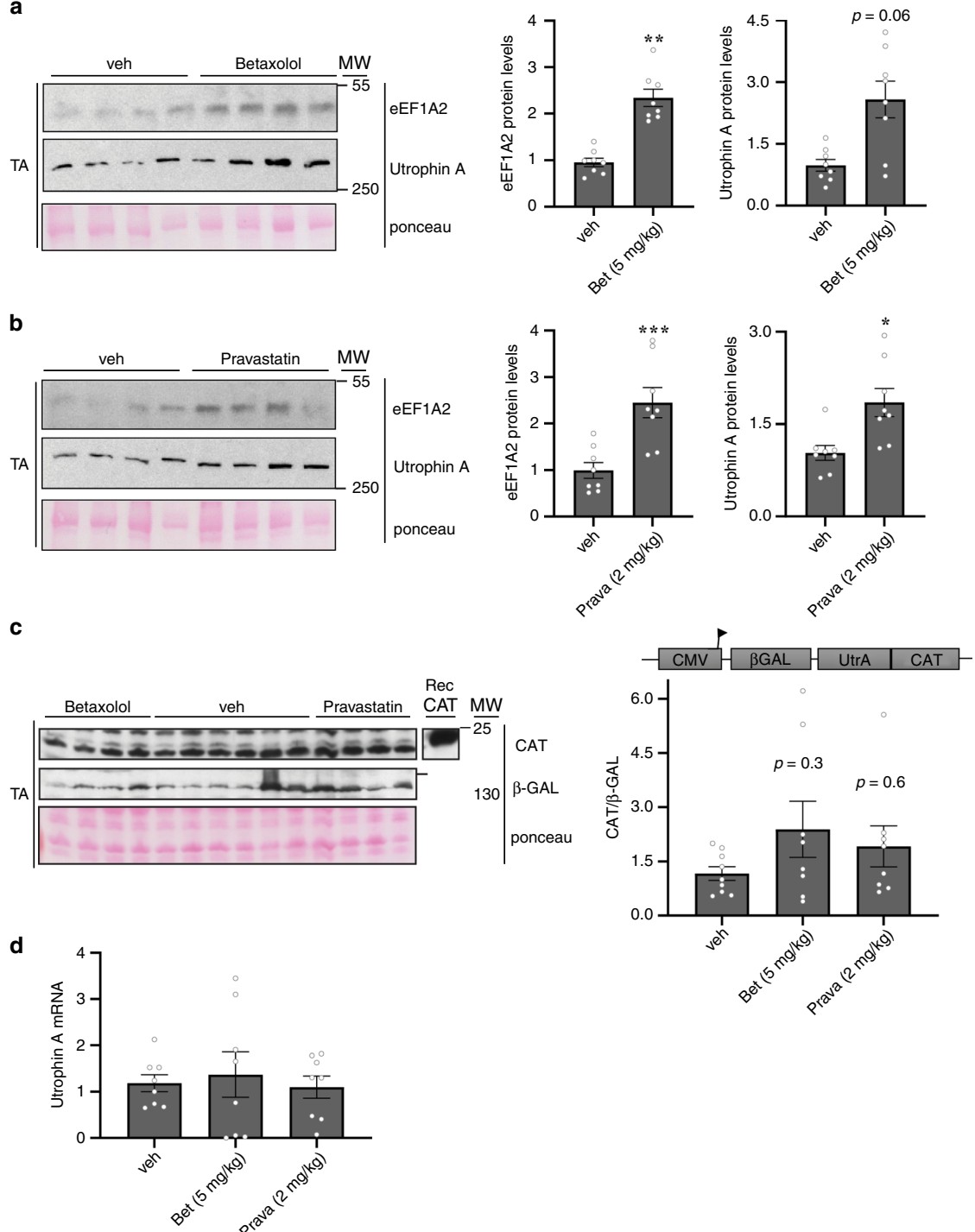

**Fig. 3 Pharmacological stimulation of utrophin A in transgenic bicistronic utrophin 5'UTR IRES harboring mice.** Transgenic mice were treated with Betaxolol (Bet) (5 mg/kg), Pravastatin (Prava) (2 mg/kg) or vehicle for 7 days. **a, b** Western blots and quantification of eEF1A2 and utrophin A protein levels normalized to ponceau using protein extracts from TA muscles from the treated transgenic mice. **c** Activation of utrophin 5'UTR IRES reporter construct after a 7-day treatment with Bet and Prava in transgenic mice. Representative western blots of CAT and β-GAL protein levels from TA muscles from the treated transgenic mice. The ratio of CAT/β-GAL level of each drug normalized to vehicle control represents the IRES activity. Recombinant CAT is used as a positive control. **d** Utrophin A mRNA levels in Bet-, Prava- or vehicle-treated transgenic mouse TA muscles, standardized to 18S. $N = 8$, error bars represent SEM. *$P < 0.05$, **$P < 0.01$, ***$P < 0.001$, vs vehicle control. Two-tailed Student's $t$-test (**a**, **b**) and one-way ANOVA, accompanied by Bonferroni post-hoc test (**c**, **d**), were performed to determine statistical differences in these experiments. Source data are provided as a Source Data file.

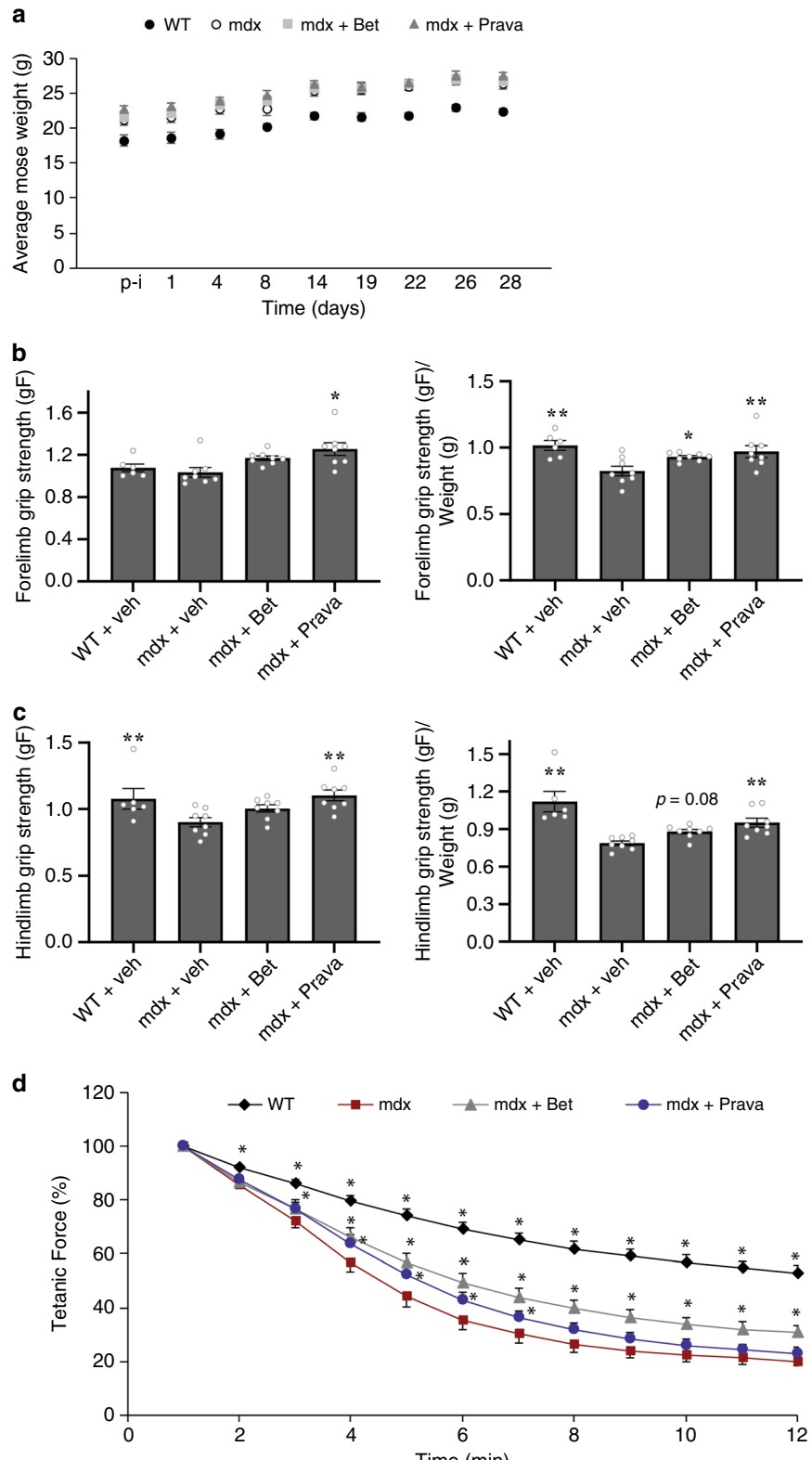

**Fig. 4 The effects of Betaxolol and Pravastatin treatment on muscle strength of mdx mice.** Mdx mice were treated with Betaxolol (Bet), Pravastatin (Prava) or vehicle for 4 weeks. **a** mdx and wild-type (WT) mouse body weight (g). **b** Forelimb and **c** Hindlimb grip strength analysis of WT, vehicle-treated, Bet- or Prava-treated mdx mice normalized or not normalized to body weight. $N = 6$ for WT and $N = 8$ for mdx + veh or treated, error bars represent SEM. *$P < 0.05$, **$P < 0.01$ vs vehicle treated mdx mice. One-way ANOVA and Bonferroni post-hoc test was performed to determine statistical differences for (**a–c**). A split-plot ANOVA was used for force drop analysis (**d**). Source data are provided as a Source Data file.

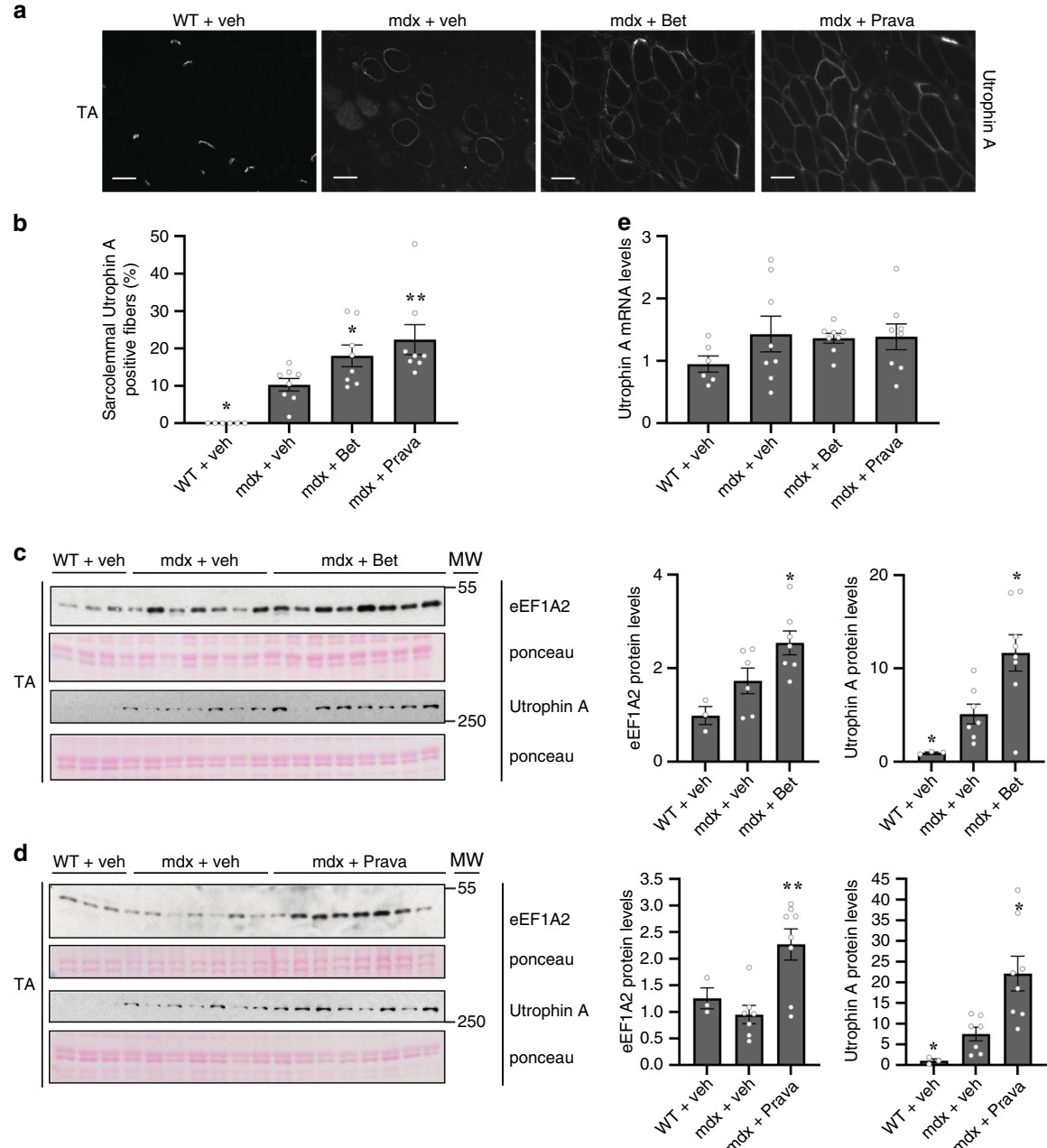

**Fig. 5 Increase of sarcolemmal localization and protein levels of utrophin A by Betaxolol and Pravastatin treatment in mdx mice. a** Representative examples of cross-sections obtained from TA muscles of wild-type (WT) mice and mdx mice treated with Betaxolol (Bet), Pravastatin (Prava) or vehicle control (saline) for 4 weeks, were immunostained with utrophin A (UTR-A) antibody. Scale bars, 50 mm. **b** Quantification of sarcolemmal utrophin A positive fibers to total muscle fibers. **c, d** Western blots and quantification of eEF1A2 and utrophin A protein levels normalized to ponceau using protein extracts from TA muscles from the treated mdx mice. **e** Utrophin A mRNA levels in Bet-, Prava- or vehicle-treated mdx mouse TA muscles, standardized to 18S. $N = 6$ for WT and $N = 8$ for mdx + veh or treated. Error bars represent SEM. *$P < 0.05$, **$P < 0.01$, vs vehicle control. One-way ANOVA and Bonferroni post-hoc test was performed to determine statistical differences. Source Data for **b–e** are provided in the Source Data File.

In addition, longitudinal sections of TA muscles treated with each drug showed increased utrophin A protein localization extending from neuromuscular junctions (NMJ) into extrasynaptic regions, as compared to vehicle control (Supplementary Fig. 2).

In addition to these localization studies, western blot analyses showed significant increases of both eEF1A2 (~1.5 and 1.8-fold; $P < 0.05$ and $P < 0.01$, one-way ANOVA) and utrophin A protein levels (~2.2 and 3.0-fold; $P < 0.05$, one-way ANOVA) in TA muscles from Betaxolol- and Pravastatin-treated mdx mice (Fig. 5c, d). In agreement with a translational induction in utrophin A expression, qRT-PCR results showed no change in

utrophin A mRNA levels in TA muscles from Betaxolol- or Pravastatin-treated mdx mice when compared to vehicle control (Fig. 5e). This further indicates that the upregulation of utrophin A following treatment with either one of these two drugs occurs through translational events.

**Betaxolol and Pravastatin improve muscle fiber integrity**. We next determined the effects of Betaxolol and Pravastatin on the morphology and integrity of dystrophic muscle fibers. To do so, we analyzed changes in central nucleation, a marker of muscle fiber regeneration[43], by performing Hematoxylin and Eosin

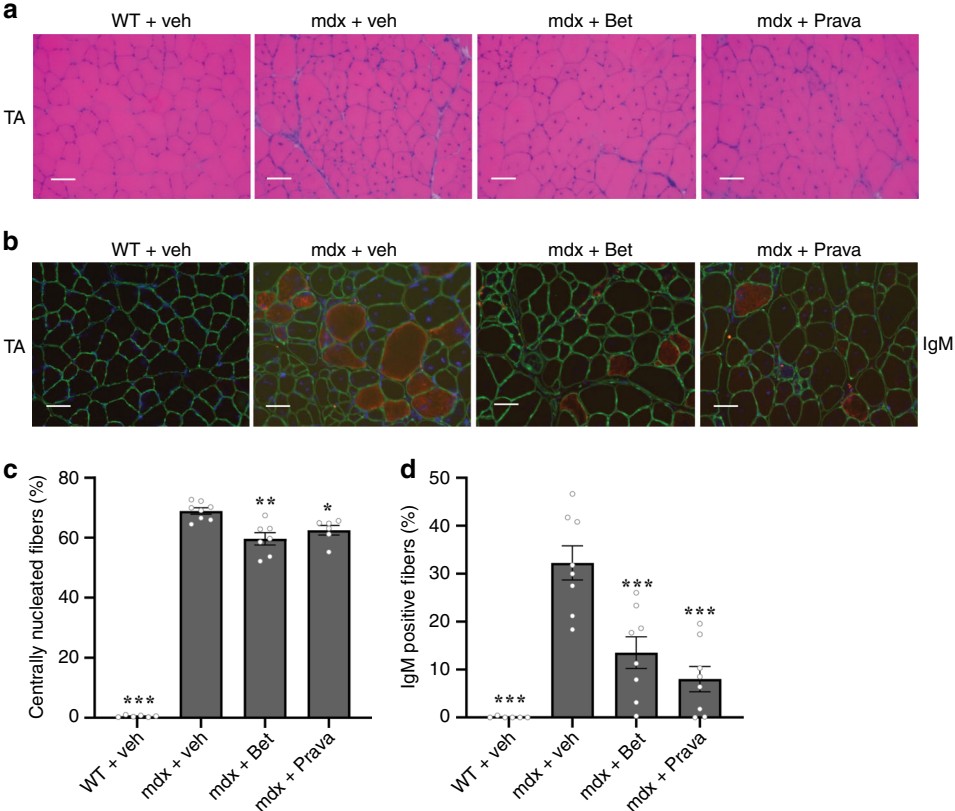

**Fig. 6 Morphological features of mdx muscle fibers treated with Betaxolol and Pravastatin. a** Representative examples of cross-sections of TA muscles from wild-type (WT) and mdx mice treated with Betaxolol (Bet), Pravastatin (Prava) or with vehicle (saline) that were stained using hematoxylin and eosin. **b** Representative examples of cross-sections of TA muscles from WT mice and mdx mice treated with Bet, Prava or vehicle that were immunostained with goat-anti-mouse IgM Alexa 594. **c** Percentage of central nucleation in TA muscle fibers. **d** Quantification of IgM positive fibers vs total muscle fibers. $N = 6$ for WT and $N = 8$ for mdx + veh or treated. Error bars represent SEM, *$P < 0.05$, **$P < 0.01$, ***$P < 0.001$, significantly different from mdx vehicle control. Scale bars, 50 mm. One-way ANOVA and Bonferroni post-hoc test was performed to determine statistical differences. Source Data for c and d are provided in the Source Data File.

staining on cryostat sections, as well as intracellular IgM staining that reflects sarcolemmal damage. Our data revealed that both drugs decreased central nucleation ($P < 0.05$ and $P < 0.01$, one-way ANOVA) (Fig. 6a, c), and induced a striking ~3-fold reduction in IgM infiltration into muscle fibers ($P < 0.001$, one-way ANOVA) (Fig. 6b, d). Altogether, these experiments show that increasing expression of utrophin A, through drug-induced activation of eEF1A2, attenuates the dystrophic pathology in mdx mice thereby illustrating the therapeutic potential of these drugs for treating DMD patients.

**Betaxolol and Pravastatin increase utrophin A in human cells**. To further determine the clinical potential of Betaxolol and Pravastatin, we treated human skeletal muscle cells (SkMC) with each drug or vehicle control for 24 h (Supplementary Fig. 3). As seen in mouse C2C12 muscle cells, utrophin A and eEF1A2 protein levels increased in human cells treated with Betaxolol and Pravastatin. These data support our findings in vitro and in vivo with mice, and further demonstrate the potential of these two drugs for treating DMD patients.

**Pravastatin fails to upregulate utrophin A in wasted mice**. To determine whether utrophin A upregulation by Pravastatin is directly dependent on eEF1A2, we performed daily treatments of eEF1A2-null and wild-type mice with either Pravastatin or vehicle control for 5 days. eEF1A2-null mice, commonly referred to as wasted mice, possess a naturally occurring ~16 kb deletion

that eliminates the first non-coding exon and regulatory promoter elements of the gene encoding eEF1A2, leading to complete ablation of its expression[44,45]. eEF1A1 is highly expressed in neuronal, cardiac and skeletal tissues during embryonic development but its expression gradually declines after birth until it becomes undetectable by day 21. By contrast, eEF1A2 expression in these tissues increases starting shortly before birth until it reaches a plateau by day 21[46]. Therefore, wasted mice show striking neuromuscular deficits starting at day 21 which leads to their death at approximately day 28[44,47]. Accordingly, we treated wasted and wild-type mice with Pravastatin starting at day 20 for five days.

Our data demonstrate that a short, 5-day treatment of wild-type mice with Pravastatin was sufficient to induce a ~2.7-fold increase ($P < 0.05$, two-tailed Student's $t$-test) in the levels of endogenous utrophin A compared to vehicle control and a trend toward an increase in eEF1A2 levels (Fig. 7a, c). Remarkably, and in complete agreement with our working hypothesis, Pravastatin treatment of eEF1A2-null mice did not cause an increase in utrophin A expression (Fig. 7b, c). These data show that Pravastatin-mediated upregulation of utrophin A is dependent on eEF1A2.

**Discussion**
In the present study, we set out to: (1) examine whether eEF1A2 regulates utrophin A expression through IRES-mediated translation; and (2) identify and characterize FDA-approved drugs that

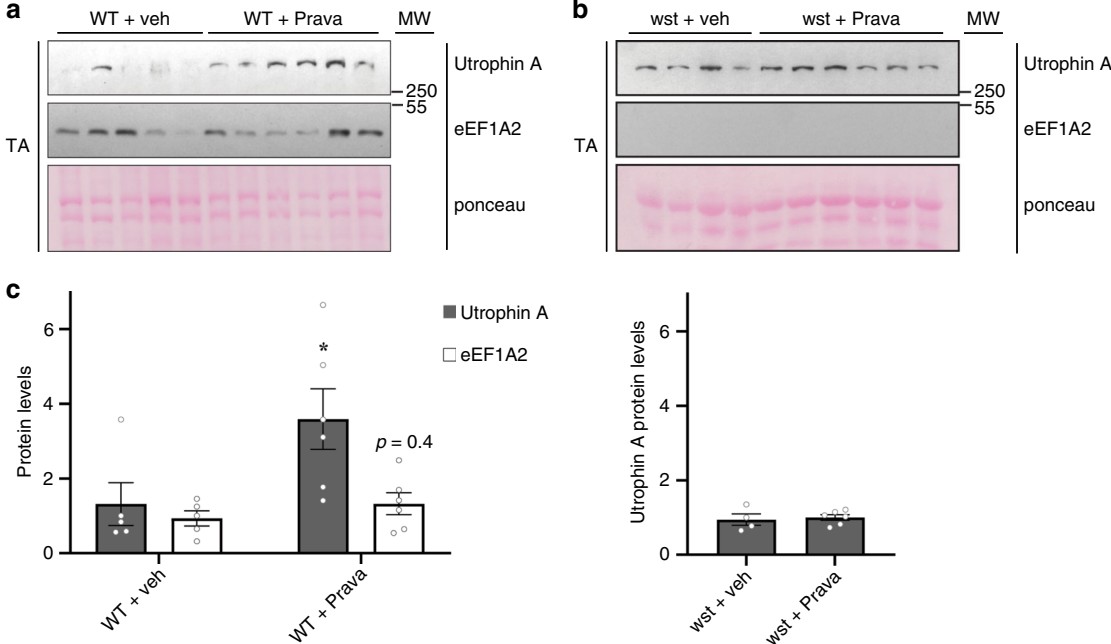

**Fig. 7 Pravastatin does not induce a utrophin A increase in eEF1A2-null mice.** eEF1A2-null mice (wasted mice—wst) and wild-type mice (WT) were treated with Pravastatin (Prava) (2 mg/kg) or saline for 5 days. **a**, **b** Western blots of utrophin A and eEF1A2 protein levels using protein extracts from TA muscles from wild-type and wasted mice treated with pravastatin or vehicle control. **c** Quantification of utrophin A and eEF1A2 protein levels normalized to ponceau in treated WT and wasted mice. $N = 4$ for WT or wst + veh and $N = 6$ for WT or wst + Prava. Error bars represent SEM, *$P < 0.05$, significantly different from vehicle control. Two-tailed Student's $t$-test was performed to determine statistical differences. Source data are provided as a Source Data file.

might offer benefits to dystrophic muscles by acting through this pathway. Through a series of complementary studies, we demonstrate that eEF1A2 directly regulates endogenous utrophin A protein expression via IRES-dependent translation. Moreover, we designed a high-throughput screen targeting eEF1A2 and identified seven classes of FDA-approved drugs able to activate IRES-dependent translation of utrophin A. Further investigation of two of these drugs, Betaxolol and Pravastatin, revealed their ability to activate this pathway in vivo and to improve the dystrophic phenotype of mdx mice. Taken together, our findings illustrate the feasibility of targeting eEF1A2/IRES-mediated translation of utrophin A with repurposed drugs as a therapeutic approach for treating DMD.

In this work, we compared endogenous levels of eEF1A2 in both slow soleus and fast EDL muscles of mdx vs wild-type mice. Previous studies have shown that utrophin A mRNA and protein are more abundant in slow muscles compared to fast ones, and that this increased expression is in part due to an enrichment of utrophin A in extrasynaptic regions[48]. Here, we did not detect a difference in endogenous eEF1A2 levels between fast or slow skeletal muscles indicating that eEF1A2 does not play a significant role in differentially regulating utrophin A expression in fast versus slow muscles. Moreover, we observed that eEF1A2 is similarly expressed in muscles of wild-type versus mdx mice. This latter finding is important as it suggests that overexpression of eEF1A2 in mdx muscles may indeed be a viable therapeutic approach for increasing endogenous utrophin A levels in dystrophic muscle fibers as we have shown here in proof-of-principle studies.

From our in-cell ELISA-based high throughput drug screen that contained 262 FDA-approved drugs, we identified 11 drugs that activated both eEF1A2 and utrophin A. We noted that some of these drugs have common functions such as anti-diabetic, anti-peptic ulcer, cholesterol-lowering and beta-adrenergic agents.

This is interesting and consistent with the fact that several anti-diabetic drugs and statins (cholesterol-lowering drugs) have been shown to improve dystrophic muscles[10,13,49,50]. In addition, specific beta-adrenergic blockers such as Carvedilol were shown to improve cardiac function in DMD patients[51]. Nonetheless, a recent study showed that the statin simvastatin induced beneficial effects in mdx mice in part by reducing oxidative stress and increasing autophagy without an effect on utrophin A expression[52,53]. FDA-approved drugs Betaxolol and Pravastatin are multifunctional drugs that may also impact autophagy. In the current study, however, a 4-week treatment with either Pravastatin or Betaxolol had no effect on the LC3A/B II to I ratios in mdx mouse muscles indicating that these drugs do not affect autophagy (Supplementary Fig. 4A, B). It is important to also note that simvastatin, a statin with high permeability in muscle, was part of the 262 drugs that were tested in our study, and that our results showed its inability to increase utrophin A protein levels as originally demonstrated[52]. Thus, in this context, the ability for statins to enter muscle fibers is not the limiting factor for eEF1A2 and utrophin A upregulation. While several drugs play common therapeutic roles, they may still activate distinct pathways explaining why only some statins affect utrophin A levels.

Following confirmation of the ability of these drugs (Table 2) to induce eEF1A2 and utrophin A expression, we decided to further investigate the five top drugs for their capacity to increase IRES-mediated translation of utrophin A. Using a cell culture system with myoblasts, we demonstrate that these five drugs all stimulate the activity of the utrophin A 5′UTR IRES reporter construct without causing any changes in utrophin A mRNA levels. This indicates that indeed these drugs promote IRES-dependent translation of utrophin A.

Based on a 7-day drug treatment of our transgenic mice harboring the bicistronic reporter construct containing the utrophin

5'UTR, we observed that the beta-androgenic blocking agent Betaxolol and the cholesterol-lowering drug Pravastatin were the most potent activators of both eEF1A2 and utrophin through its 5'UTR IRES in vivo. Furthermore, a 4-week treatment of mdx mice with these drugs elicited significant improvements of the dystrophic phenotype that include increases in muscle strength and amelioration of muscle fiber morphology and sarcolemmal integrity. In order to confirm that utrophin A upregulation was dependent upon eEF1A2 activity, we also treated eEF1A2-null (wasted) mice with Pravastatin and show that utrophin A upregulation, as seen in treated wild-type mice, was completely abolished in muscles from wasted mice. Altogether, these experiments show that the ITAF eEF1A2 is able to regulate utrophin A IRES-dependent translation and that drugs such as Betaxolol or Pravastatin can modulate utrophin A levels via eEF1A2, thereby revealing their potential as relevant repurposed agents for treating DMD.

There are two isoforms of eEF1A: eEF1A1 and eEF1A2, with both isoforms playing similar roles in translation elongation[54,55]. Over the past few years, however, a number of non-canonical roles have been identified for eEF1A, some of which appear to be specific to either 1A1 or 1A2 variants. For example, in terminally-differentiated myotubes eEF1A1 was shown to promote apoptosis, whereas eEF1A2 played an anti-apoptotic role[56], suggesting clear distinct roles for the two isoforms. Furthermore, eEF1A1 and eEF1A2 are differentially expressed in tissues. In contrast with eEF1A1 which is generally expressed ubiquitously, eEF1A2 is preferentially expressed in skeletal muscle, heart and brain tissues[57,58], suggesting that eEF1A2 plays a distinct role in muscle.

Our current findings showing the role of eEF1A2 as an ITAF are in agreement with the recent demonstration that another elongation factor, eEF2, acts as an ITAF to regulate IRES-mediated translation of XIAP and FGF2 mRNAs[59]. Currently, it is unknown how elongation factors could act as ITAFs to regulate IRES-mediated translation of specific mRNAs. However, it is possible that they may interact with tRNA-like structural elements found in IRESs given the main role of elongation factors in shuttling aminoacylated tRNAs to the ribosomal A site, enabling the continuation of protein synthesis[60]. In addition, eEF1A binds a variety of RNA structures usually located at the 3'UTR of viral genomes. In fact, eEF1A has been found to interact with TYMV tRNA-like structures and the stem-loop structures[61] at the 3'UTR of tombusvirus genomic RNA[62] as well as the West Nile virus genomic RNA[63]. However, there is no common sequence between the binding sites of these RNAs, and it is thus unclear if the utrophin A IRES contains a similar structure.

As an ITAF, it would be unlikely that eEF1A2 would enhance the activity of other cellular IRESs. IRESs are not defined by a consensus sequence or RNA structure[64]. Previous reports also demonstrate that the proteins controlling IRES-dependent translation initiation are modulated by their subcellular localization[65]. However, the regulation of protein translation by IRESs are still largely misunderstood and requires additional studies. Our laboratory has previously shown that eEF1A2 is increased in cardiotoxin-treated skeletal muscle cells which stimulated expression of a bicistronic construct containing utrophin A's IRES. By contrast, our laboratory also showed that the expression of a control bicistronic vector containing a XIAP IRES site is not stimulated under these same conditions[31,32]. Together, these findings demonstrate that eEF1A2 is not able to stimulate all IRESs. However, due to the structural similarities of utrophin A and dystrophin it would be interesting to determine whether eEF1A2 also regulates the activity of the recently described IRES found in dystrophin mRNAs[33].

In summary, our study shows that eEF1A2 directly regulates IRES-mediated translation of utrophin A. Moreover, our high-throughput drug screen identified specific classes of FDA-approved drugs able to increase utrophin A through eEF1A2 and IRES-dependent translation with at least two of them being able to significantly improve the dystrophic phenotype of mdx mice. Our work thus reveals the therapeutic relevance of eEF1A2 as a target for pharmacological interventions in DMD and shows the feasibility of using repurposed drugs to activate this pathway for treating this neuromuscular disorder.

## Methods

**In-cell ELISA high-throughput drug screen.** An ELISA-based high-throughput limited drug screen was designed using an In-Cell ELISA Colorimetric Detection Kit (Thermo Fisher Scientific, Massachusetts, USA). For this, a total of 262 FDA-approved drugs were aliquoted in 384-well microplates derived from the Screenwell FDA-approved drug library V2 (Enzo Life Sciences—Cederlane—Ontario, Canada). Each drug dose for the initial drug screen was based on doses used in the clinic by patients as listed in the Drug Bank database[66]. C2C12 myoblasts (American Type Culture Collection, Virginia, US, CRL-1772™) were grown in each well and treated with these drugs or vehicle control for 24 h. Following treatment, antibodies targeting eEF1A2 (1:1000, provided by Dr. Abbott) or utrophin A (1:500; Novocastra, Leica biosystems, Concord, ON, Canada, NCL-DRP2) and HRP-conjugated IgG secondary antibodies (Jackson Immuno Research, Bar Harbor, USA, 111-035-033 and AP124P) were used to detect protein expression levels. Absorbance levels were determined with a Synergy H1 microplate reader. Note that the absorbance levels are standardized to total cell number by using a whole-cell stain in order to control for variation in cell proliferation.

**Reporter assays.** C2C12 skeletal muscle myoblasts (ATCC) were transfected with one bicistronic construct, either the control CMV/βGAL/CAT or the CMV/βGAL/UtrA/CAT containing the utrophin A 5'UTR. The next day, cells were treated with various drugs or vehicle control for 24 h. Reporter activity from cell lysates was assessed by using the CAT and β-GAL ELISA reporter assay kits (Roche, QC, Canada) and a Synergy H1 microplate reader. The activity of utrophin A IRES was determined by establishing a ratio of CAT to β-GAL activity. It is important to note that extensive control experiments have been previously performed to ensure that the bicistronic mRNA does not undergo aberrant splicing, while also showing that the utrophin A 5'-UTR does not contain cryptic promoter activity[31,32].

For in vivo analysis, proteins from transgenic mouse TA muscles were extracted with reporter lysis buffer using the β-GAL enzyme assay system according to the manufacturer's instructions (Promega, Wisconsin, USA). The protein concentration was determined using the bicinchoninic acid (BCA) assay. Protein samples were diluted to a final concentration of 4 mg/ml prior to performing the reporter assays. β-GAL enzymatic assays were performed using the β-GAL enzyme assay system as recommended by the manufacturer (Promega). To measure CAT activity, we analyzed the conversion of chloramphenicol to butyryl-chloramphenicol by incorporation of [14C] butyryl coenzyme A[32]. Background levels for the reporter assay were determined by analyzing reporter activity in tissues from mice not harboring a transgene.

**In vitro drug treatment.** C2C12 myoblasts (ATCC) were treated with 11 drugs: Acarbose, Betaxolol, Labetalol-HCl, Telbivudine, Pravastatin-Na, Olsalazine-Na, Lovastatin, Nizatidine, Propylthiouracil, Hydrochlorothiazide and Rosiglitazone at concentrations indicated (Supplementary Fig. 1). 24-h later, cells were harvested in urea buffer and subjected to western blot analysis. Human Skeletal Muscle Cells (SkMC) (Lonza, NJ, USA, CC-2561), grown in SkBM™-2 (Lonza, CC-3245) supplemented with SkGM™-2 SingleQuots™ supplements (Lonza, CC-3244), were treated with Betaxolol (0.15 μM and 1 μM) and Pravastatin (20 nM and 80 nM) for 24 h. Lysates were subjected to western blot analysis as described below.

**Animal strains and experiments.** The animal studies were performed under ethical regulation of the University of Ottawa Animal Care Committee with the use of approved animal protocol # CMM-2285, working within University of Ottawa animal facilities and with veterinary as well as animal care staff. The University of Ottawa is certified by the Canadian Council on Animal Care and has an Animal Welfare Assurance with the US Public Health Service. In vivo experiments were performed using C57BL/10ScSn-Dmdmdx/J (mdx) mice, C57BL/10 (wild-type) mice (The Jackson Laboratory, Bar Harbor, USA), B6C3Fe a/a-EEF1A2wst/J mice and B6C3Fe a/a wild-type mice (The Jackson Laboratory—JAX stock #000182), as well as the transgenic mice harboring either a CMV/bGAL/CAT or a CMV/bGAL/UtrA/CAT bicistronic reporter transgene[32]. These mice were maintained in the Animal Care and Veterinary Service at the University of Ottawa. All animal work has complied with all relevant ethical regulations. There were 6–8 mice in each animal group. Our sample size was calculated based on Charan et al.[67].

Ectopic overexpression of eEF1A2 in WT and mdx mice were performed using eEF1A2 expression constructs (MYC-HIS360-eEF1A2-pcDNA) or pcDNA3.1

control. The plasmids were directly electroporated into either one of the two TA muscles while the animals were under anesthesia. Seven days after electrotransfer, mice were euthanized and TA muscles were dissected for further analysis.

**In vivo drug treatment**. CMV/bGAL/UtrA/CAT transgenic mice were treated daily with either Betaxolol (5 mg/kg) (Selleckchem, Houston, Texas, USA), Pravastatin-Na (2 mg/kg) (Santa-Cruz, Dallas, Texas, USA) or vehicle control (saline) for 7-days by intraperitoneal injection (IP). B6C3Fe *a/a*-EEF1A2[wst]/J mice and B6C3Fe *a/a* wild-type were treated with Pravastatin-Na (2 mg/kg) or vehicle control (saline) for 5-days by IP injection. A shorter treatment time was performed with these mice due to their short life expectancy of approximately 28 days[68]. Both drugs were dissolved in sterile saline prior to each treatment. Six-week-old mdx or wild-type mice were treated daily with Betaxolol (5 mg/kg/day), Pravastatin-Na (2 mg/kg/day) or vehicle control (saline) by IP injection for 4-weeks, a treatment period regularly performed in our laboratory[10,13,14,69]. Muscles were then dissected from euthanized mice and either flash frozen in liquid nitrogen or embedded in Optimum Cutting Temperature compound (OCT) and frozen in melting iso-pentane cooled with liquid nitrogen.

**Forelimb grip strength**. Forelimb grip strength analysis was performed on the final day of drug treatments and was evaluated with the use of a digital force gauge, Chatillon DFE II (Columbus Instruments, Columbus, USA) and a grid. The mice were first acclimatized to the work area for 30 min. They were then permitted to grip the grid attached to the digital gauge and pulled horizontally away from the bar in a constant motion, until release of the grid. The process was repeated six times per mouse accompanied by a 30 s rest time between each measurement. The value of the maximal peak force was recorded (gF). The grip strength measurements were conducted by the same investigator in order to limit variability and were performed in a random order. The investigator performing the measurements was blinded as to the treatment group of each individual mouse upon testing.

**Ex vivo eccentric contractions and force drop analysis**. After 4–6 weeks, wild-type as well as Betaxolol-, Pravastatin- and vehicle-treated mdx mice were euthanized. The EDL muscle was dissected and attached at one end of a Dual mode lever system (model 300C, Aurora Scientific, Aurora, Canada) to measure force and to lengthen muscle, and the other end to a fixed rod. Throughout the experiment, the muscle was submerged in a saline solution containing (in mM): 118.5 NaCl, 4.7 KCl, 2.4 $CaCl_2$, 3.1 $MgCl_2$, 25 $NaHCO_3$, 2 $NaH_2PO_4$, 5.5 D-glucose, 95% $O_2$–5% $CO_2$ (to maintain a pH of 7.4), with a flow rate of 15 mL/min at room temperature. Adjustments of the muscle length were performed in order to get maximal force output. Five maximal tetanic contractions (400 ms train duration, 10 V, 0.3 ms square pulse, 200 Hz) were elicited to determine muscle contractile kinetics. These contractions were executed every 100 s, followed by 12 eccentric contractions every 120 s (700 ms train duration, 10 V, 0.3 ms square pulse, 200 Hz) Eccentric contractions were elicited by subjecting muscles to a 10% lengthening at a velocity of 0.5 Le/s throughout the last 200 ms. Electrical stimulation was generated across two platinum wires (positioned above and below muscles 4 mm apart) using a Grass stimulator (model S88X, Grass Technologies, West Warwick, USA). A Keithley data acquisition board (model KPCI-3104, Cleveland, USA) was used to detect the force at a sample rate of 5 KHz. Force drop data were expressed as mean ± standard error (S.E.).

**Western blotting**. C2C12 skeletal muscle cells or mouse muscle tissues were homogenized in urea buffer supplemented with protease inhibitor (Roche). A total of 5–20 μg of protein extracts were resolved on either a 7% SDS-PAGE gel for utrophin A and eEF1A2 analyses, or 10% SDS PAGE gels for CAT and β-GAL analyses. Proteins were transferred overnight at 4 °C onto nitrocellulose membrane (Bio-Rad, Mississauga, ON, Canada, 0.45 μm). Membranes were subsequently washed 4 times with 1× PBS-T (1xPBS, 0.2% Tween) and blocked for 1 h with a 5% skim milk in PBS-T solution. Membranes were incubated with primary antibodies directed against utrophin A (1:500; Novocastra, NCL-DRP2), eEF1A2 (1:1000; provided by Dr. Abbott and Abcam, Toronto, ON, Canada), myc tag (1:1000, Abcam, ab9132), CAT (1:1000, Abcam, ab50151), β-GAL (1:1000, Abcam, ab616), LC3A/B (1:1000; Cell Signaling, Danvers, MA, USA, 12741S) and β-actin (1:10,000; Santa Cruz, sc-47778). Blots were probed with appropriate HRP-conjugated IgG secondary antibodies (Jackson ImmunoResearch, 111-035-033, AP124P and 705-035-003). Protein detection was performed by using ECL reagent (Perkin Elmer, Waltham, MA, USA). The films were quantified using ImageJ (NIH version 1.0) and/or Image Lab. Uncropped scans are available in the Source Data file.

**RNA isolation and qRT-PCR**. Total RNA was isolated from C2C12 treated cells as well as muscle tissues from wild-type and mdx mice using TRIzol reagent (Invitrogen, Carlsbad, California, USA). TRIzol extracted RNA was treated for 1 h with DNAse I (Invitrogen, Carlsbad, CA, USA) followed by reverse transcription (RT) using an RT reaction mixture (5 mM MgCl2, 1× PCR buffer, 1 mM dNTP, 1 U/ml RNase inhibitor, 5 U/ml Moloney murine leukemia virus reverse transcriptase and 2.5 mM random hexamers) (Applied Biosystems, CA, USA). qPCR was performed on an MX3005p real-time PCR system (Stratagene, La Jolla, CA, USA) using the QuantiTect SYBR Green PCR kit (QIAGEN, Valencia, CA, USA). Utrophin A

amplification and house keeping gene 18S ribosomal subunit was performed in duplicates with the following primer sequences: utrophin A—forward 5′-ATCTTGTCGGGCTTTCCAC-3′ and reverse 5′-ATCCAAAGGCTTTCCCA-GAT-3′, 18S Ribosomal—forward 5′-CGCCGCTAGAGGTGAAATC-3′ and reverse 5′-CCAGTCGGCATCGTTTATGG-3′,

**Immunofluorescence**. Immunofluorescence experiments were performed using ten μmeter cross-sections or longitudinal sections of wild-type, Betaxolol-, Pravastatin- or saline-treated mdx mouse TA muscles. Sections were prepared for immuno-fluorescence analysis using the M.O.M Immunodetection kit (Vector Laboratories, Burlington, ON, Canada). The sections were incubated with primary antibody directed against utrophin A (1:200, Novacastra, NCL-DRP2) and Texas Red-conjugated Streptavidin antibody (1:500; Vector laboratories, SA-5006) or against a FITC-conjugated IgM anti-mouse secondary antibody (1:400; Sigma-Aldrich, Oakville, Canada, F9259). All muscle tissue sections were co-stained with a rabbit laminin antibody (1:800; Sigma-Aldrich, Oakville, Canada, L9393) along with goat anti-rabbit Alexa Fluor 488 IgG secondary antibody (1:500; ThermoFisher Scientific, Massachusetts, USA, A-11034) to highlight the sarcolemma and with Alexa 488-conjugated Bungarotoxin (1:500; ThermoFisher Scientific, B13422) to stain NMJs. The slides were mounted with Vectashield containing DAPI staining (Vector Laboratories, Burlington, ON, Canada) and visualized using a Zeiss Axioskop-2 microscope. Quantification was accomplished using Image J (NIH version 1.0).

TA muscle cross-sections were stained with Hematoxylin and Eosin dyes. Sections were dehydrated using 70%, 90%, and 100% ethanol solutions and washed with toluene. The sections were mounted using Permount (Fisher Scientific, Ottawa, Canada) and visualized using an epifluorescent EVOS FLAuto2 inverted microscope. Percentage of central nucleation was determined by counting the total number of muscle fibers and the number of centrally nucleated muscle fibers from 6 to 8 cross-sectional views using the Northern Eclipse Software (NES, EMPIX Imaging, Mississauga, Ontario, Canada).

**Statistical analysis**. The data were analyzed using two-tailed Student's *t*-test and one-way ANOVAs (Analysis of Variance) with Bonferroni post-hoc tests. Error bars represent standard error of the mean (SEM). Statistical analysis was done with Graph Pad prism 6 (Prism Software, La Jolla, CA, USA). The level of significance was set at $P \leq 0.05$.

Split-plot ANOVA designs were used for eccentric contractions to determine statistical differences[70]. Comparisons between mouse groups involved muscles from different mice and thus the data were independent from one another; in this case, the S.E. for statistical differences was the population S.E. (whole plot). Comparisons between eccentric contractions (and force-frequency relationship) involved data from the same muscles, and thus data were not independent from one another; in this case, the S.E. for statistical differences was the S.E. between muscles (split plot). Calculations were made using the GLM (General Linear Model) procedures of the Statistical Analysis Software version 9.4 (SAS Institute Inc., Cary, NC, USA). When a main effect or an interaction was significant, the least square difference (L.S.D.) was used to locate the significant differences. The word "significant" refers only to a statistical difference ($P < 0.05$).

## Data availability

Supplementary Figs. 1–4 are provided as a Source Data file. All data generated and analyzed during this study are included in this article and its Supplementary Information file. All data is available from the corresponding author upon reasonable request.

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

## Acknowledgements

The authors are grateful to John Lunde for expert technical assistance and Dr. C.M. Abbott for providing the eEF1A2 antibody used in this study. This work was supported by grants from Jesse's Journey, the Association Française contre les Myopathies, the Muscular Dystrophy Association (USA) and the Canadian Institutes of Health Research. C.P. benefited from CNMD Scholarships in Translational Research and the Ontario Graduate Scholarship throughout this work.

## Author contributions

Conceived and designed the experiments: C.P. and B.J.J. Performed the experiments: C.P., N.A., A.C., H.A., and M.T. Analyzed the data: C.P. and N.A. Provided expertise: L.M.B., J.M.R., A.M., and M.H. Contributed reagents/materials/analysis tools: J.V., A.M., and M.H. Wrote the paper: C.P. and B.J.J.

## Competing interests

The authors declare no competing interests.
