## [Peer Review File · Nature Communications]

Reviewers' Comments:

Reviewer #1:

Remarks to the Author:

The manuscript by Peladeau et al analyzes the effect of 262 FDA approved drugs on the expression of utrophin A, a protein that behaves as the autosomal homologue of dystrophin. Low levels of dystrophin in muscle are responsible for the Duchenne muscular dystrophy (DMD). Hence, compounds upregulating utrophin A could be important tools for therapeutic intervention of DMD.

In this manuscript the authors first determine that overexpression of eEF1A2 stimulates the expression of utrophin A in wild type (C5BL/19) and mdx (C5BL/19/10ScSnDmdmdx/J9) mouse muscles. Stimulation of utrophin A depends on an IRES element present on the 5' UTR of the mRNA, previously reported by these authors. Then, they study how different FDA-approved compounds affect eEF1A and Utrophin A protein levels using an Elisa-based screen in C2C12 cells. From these data they select 5 drugs that appear to have stimulatory effect. The study of these drugs regarding the potential effect on utrophin A (Utr A) IRES activity suggested that Betaxolol and Pravastatin activate the expression of utrophin A protein through eEF1A2. Finally, they analyze how these drugs can enhance muscle strength of mdx mice. They conclude that Betaxolol and Pravastatin treatment of mdx mice improve the integrity of sick muscle fibers in eEF1A2-dependent manner. Overall, the manuscript addresses an important point, which is of great interest. However, there are serious concerns that need to be addressed to support the conclusions.

1. Figure 1C. The expression levels of eEF1A2 are not shown. This is required to demonstrate that enhancement of UtrA IRES depends on eEF1A levels, as they stated in the figure legend. The WB image shown in Fig 1B corresponds to WT and mdx muscles. Data for TA (Bgal/UtrA/CAT) mice used in panel C are not shown.
2. The authors should show if the enhancement of protein levels measured by Elisa in Figure 2 could be proven by an alternative method. There are too big error bars in the histogram for some compounds. It could be wise to show box plots to visualize differences among the different compounds and the vehicle (veh). Moreover, there is no good correlation between stimulation of eEF1A and Utrophin A by the 11 drugs selected as hits.
3. Figure 3C. There is too much variability in the amount of total protein (ponceau staining) in the four vehicle samples. Indeed, samples with lower amounts of ponceau staining also have lower intensity of CAT and B-gal. Therefore, the veh data invalidate the conclusions for the drug-treated TA mice.
4. The representation used in Figure 4 is confusing. The control animals (for instance, WT + saline) should be always the same, such that the reader can see the effects of the drugs-treated or the saline-treated mdx animals. The relative small differences among the values shown in panels B and C, which includes two different comparisons for veh-treated mdx mice and veh-treated WT animals, hamper the visualization of this critical experiment.
5. In the legend of Figure 5 authors state the eEF1A and utrophin A expression levels were normalized to GAPDH. As far as I can see, this is not shown on the manuscript.
6. In figure 7, the measurement of utrophin A in eEF1A2-null mice is also controversial. Although treatment of WT animals is consistent with previous figures, the three samples wst + Prava differ in utrophin A level compared to the 3 samples wst + veh. There is no good agreement between the images and the histograms.
7. Given the results obtained in this manuscript for the activation of utrophin A IRES, it will be interesting to show whether eEF1A would also enhance the activity of other cellular IRESs. This

could be relevant for IRES-dependent expression of critical proteins, such as dystrophin.

8. The discussion about the mechanism of eEF1a-dependent activation is too vague. The authors refer to tRNA-like structural elements in IRES elements. Which ones? Is there a tRNA-like in the 5' UTR of utrophin A? And to what extent it is known that these are t-RNAlike functional elements?

Reviewer #2:

Remarks to the Author:

This paper reports the effect of FDA approved drugs on the levels of utrophin A in mdx mice and proposes that these data provide support for a utrophin based strategy for the therapy of Duchenne muscular dystrophy. The authors build on their previous observations that the elongation factor eEF1A2 mediates the induction of utrophin through its interaction with the IRES element in the 5' end of the gene and thereby mediates translation. This will be of major interest to the field as this approach to DMD therapy would be applicable to all patients all over the world. Importantly, the data demonstrate that although utrophin A is expressed highly in slow muscle fibres compared to fast ones, the levels of eEF1A2 are similar making this a potential therapeutic pathway for increasing utrophin A levels in all skeletal muscle. Similar levels are found in mdx and wild type mice. They demonstrate that the effects seen are mediated through eEF1A2 as treatment of eEF1A2 null mice do not show increases in utrophin protein.

These are potentially very important results for the therapy of DMD and the quality of the data is very good. However, there are a few aspects that need to be addressed.

They used a C2C12 muscle cell line assay to screen FDA approved drugs and determined 6 classes of molecules which increase utrophin protein levels. The results show that this shows up to 2 fold increases in utrophin protein. It would be interesting to see the dose response curves. No information is given about how they determined the optimum dose or even if they are presenting the optimum dose. They just say "clinically relevant dose".

It would also be essential to demonstrate that this effect is seen DMD patient cell lines to demonstrate that the same mechanism operates in human cells and therefore can be taken forward to the clinic.

The paper presents foregrip strength data as an assessment of the effect of 2 of these drugs on muscle after injection into mdx mice. The results show a significant positive effect of the drugs. I wonder why they did not use force drop for their assessments of muscle function as this is the standard test used to assess DMD therapies in the mdx mouse worldwide. It is thus difficult to assess their functional rescue in the mdx mouse compared to other therapeutic approaches for DMD.

Does the utrophin protein at the sarcolemma spread away from the NMJ? This could be shown in longitudinal sections of the treated mdx mouse muscle.

Reviewer #3:

Remarks to the Author:

In the manuscript entitled "Identification of novel therapeutics that target eEF1A2 and upregulate utrophin A translation in dystrophic muscles" Christine Péladeau et al show that overexpression of eEF1A2 in mdx mouse TA muscles results in an increase in IRES-mediated translation of utrophin A and then identify 11 FDA-approved drugs that increase eEF1A2 and utrophin A protein levels in skeletal muscle cell cultures. They then show that treatment of mdx mice with pravastatin and betaxolol improved the dystrophic phenotype and that pravastatin-induced utrophin A upregulation

is attenuated in eEF1A2-null mice. The authors claim that these findings demonstrate that eEF1A2-dependent IRES-mediated translation of utrophin A is an important mechanism of regulating utrophin A expression and suggest that pravastatin and betaxolol could be utilized to treat DMD.

Overall the authors should be commended on their discoveries and for preparing a manuscript that is clearly written and easy to follow. The identification of the eEF1A2 link to utrophin A translation appears to be conceptually novel as do the effects of the 11 identified drugs reported in this manuscript.

However, there are a few questions related to data interpretation and the level of scientific rigor applied to some studies, as well as concerns related to statistical methods that should be addressed.

- 1.) The statistical method(s) used in each figure two-tailed student t-tests vs. one-way ANOVAs should be included in the legend
- 2.) It is encouraging that the effects of prava are observed at clinically relevant exposures; however, can the authors explain the potential discrepancy between the finding for pravastatin treatment reported in the supplementary figure (difference only seen at 200 nM?) vs. 50nM in the primary screen
- 3.) Figure 5, are the differences between WT and mdX veh mice statistically different? Can the authors address why eEF1A2 levels appear to increase in the bet experiment and decrease in the prava experiment?
- 4.) Figure 6, Figure legend appears to be out of order
- 5.) Figure 7, the levels of eEF1A2 should also be reported
- 6.) What were the number of animals treated in each treatment group and how can we be sure that the study was sufficiently powered to detect an effect?
- 7.) It is unclear why only pravastatin was validated in the eEF1A2-null mouse. Did the authors test betaxolol in this model? And what were the results?
- 8.) Overall there is a lack of mechanistic insight into how these two drugs are mediating this effect. Why is pravastatin different/unique? Other statins are structurally related and demonstrated greater permeability in muscle.
- 9.) Statins modify circulating lipids levels, cholesterol disposition, and inflammatory processes. Could any of these parameters have changed in the reported models? Some in vivo characterization should be conducted. Also, statins have been shown to increase autophagy in similar models, the authors should either interrogate this further, or address in more detail in the discussion.

Answer to Reviewers' Comments

Reviewer #1

The manuscript by Peladeau et al analyzes the effect of 262 FDA approved drugs on the expression of utrophin A, a protein that behaves as the autosomal homologue of dystrophin. Low levels of dystrophin in muscle are responsible for the Duchenne muscular dystrophy (DMD). Hence, compounds upregulating utrophin A could be important tools for therapeutic intervention of DMD.

In this manuscript the authors first determine that overexpression of eEF1A2 stimulates the expression of utrophin A in wild type (C5BL/19) and mdx (C5BL/19/10ScSnDmdmdx/J9) mouse muscles. Stimulation of utrophin A depends on an IRES element present on the 5'UTR of the mRNA, previously reported by these authors. Then, they study how different FDA-approved compounds affect eEF1A and Utrophin A protein levels using an Elisa-based screen in C2C12 cells. From these data they select 5 drugs that appear to have stimulatory effect. The study of these drugs regarding the potential effect on utrophin A (Utr A) IRES activity suggested that Betaxolol and Pravastatin activate the expression of utrophin A protein through eEF1A2. Finally, they analyze how these drugs can enhance muscle strength of mdx mice. They conclude that Betaxolol and Pravastatin treatment of mdx mice improve the integrity of sick muscle fibers in eEF1A2-dependent manner. Overall, the manuscript addresses an important point, which is of great interest. However, there are serious concerns that need to be addressed to support the conclusions.

We would like to thank the reviewer for carefully reading the manuscript and providing important insights and suggestions that increased the quality of our work. We were delighted to read the positive comments from the reviewer on the interest and potential of our work.

1. Figure 1C. The expression levels of eEF1A2 are not shown. This is required to demonstrate that enhancement of UtrA IRES depends on eEF1A levels, as they stated in the figure legend. The WB image shown in Fig 1B corresponds to WT and mdx muscles. Data for TA (Bgal/UtrA/CAT) mice used in panel C are not shown.

This is an excellent point. We have added representative western blots on the right side of figure 1C demonstrating expression levels of the myc tag containing pcDNA-eEF1A2 construct and eEF1A2 expression levels in the TA of transgenic mice.

2. The authors should show if the enhancement of protein levels measured by Elisa in Figure 2 could be proven by an alternative method. There are too big error bars in the histogram for some compounds. It could be wise to show box plots to visualize differences among the different compounds and the vehicle (veh). Moreover, there is no good correlation between stimulation of eEF1A and Utrophin A by the 11 drugs selected as hits.

To address this important point, we have included, in supplemental figure 1, a series of immunoblotting results demonstrating that the protein level increase of eEF1A2 and utrophin A detected by ELISA, following a 24h treatment of skeletal muscle cells by the top FDA-approved drugs, is also reproducible by western blot.

As recommended, we presented the quantification of eEF1A2 and utrophin A protein levels from the treatment of the top drugs as a box plot (see below). Given that the drug testing was done in triplicates, we feel that the box plot did not effectively visualize the distribution of values and differences (boxes are only formed when you have an increased n value). We believe therefore that the grouped bar graph is most representative of our findings. However, if the reviewer insists, we will gladly change the bar graph to a box plot.

3. Figure 3C. There is too much variability in the amount of total protein (ponceau staining) in the four vehicle samples. Indeed, samples with lower amounts of ponceau staining also have lower intensity of CAT and B-gal. Therefore, the veh data invalidate the conclusions for the drug-treated TA mice.

Thank you for this suggestion, we have added new representative western blots of CAT, β -gal and ponceau with equal loading (see figure 3c)

4. The representation used in Figure 4 is confusing. The control animals (for instance, WT + saline) should be always the same, such that the reader can see the effects of the drugs-treated or the saline-treated mdx animals. The relative small differences among the values shown in panels B and C, which includes two different comparisons for veh-treated mdx mice and veh-treated WT animals, hamper the visualization of this critical experiment.

This is a good point. As suggested, we reanalyzed the grip strength data from figure 4b,c and normalized each group to WT in order to make a clear comparison between groups.

5. In the legend of Figure 5 authors state the eEF1A and utrophin A expression levels were normalized to GAPDH. As far as I can see, this is not shown on the manuscript.

We apologize for this oversight. We have removed the statement regarding GAPDH from the manuscript and have normalized to Ponceau (see figure 5c and d).

6. In figure 7, the measurement of utrophin A in eEF1A2-null mice is also controversial. Although treatment of WT animals is consistent with previous figures, the three samples wst + Prava differ in utrophin A level compared to the 3 samples wst + veh. There is no good agreement between the images and the histograms.

This is an excellent point. Accordingly, we have doubled the number of samples of both wst + Prava and wst + veh and further confirm that there is no significant difference in utrophin A expression levels between both groups (see figure 7b)

7. Given the results obtained in this manuscript for the activation of utrophin A IRES, it will be interesting to show whether eEF1A would also enhance the activity of other cellular IRESs. This could be relevant for IRES-dependent expression of critical proteins, such as dystrophin.

Our lab has previously shown that eEF1A2 is increased in cardiotoxin-treated skeletal muscle cells which stimulates expression of a bicistronic construct containing utrophin A's IRES. However, our laboratory showed that the expression of a control bicistronic vector containing a XIAP IRES site is not stimulated under these same conditions^{1,2}. Thus, this demonstrates that eEF1A2 can't stimulate all IRESs. We have added some text in the discussion to address this point (p. 25, lines 467-475 and p.26 lines 476-478)

8. The discussion about the mechanism of eEF1a-dependent activation is too vague. The authors refer to tRNA-like structural elements in IRES elements. Which ones? Is there a tRNA-like in the 5'UTR of utrophin A? And to what extent it is known that these are t-RNA like functional elements?

Thank you for this recommendation. We have now elaborated on the tRNA-like structures bound by eEF1a in the discussion (see p.25, lines 461-466).

Reviewer #2

This paper reports the effect of FDA approved drugs on the levels of utrophin A in mdx mice and proposes that these data provide support for a utrophin based strategy for the therapy of Duchenne muscular dystrophy. The authors build on their previous observations that the elongation factor eEF1A2 mediates the induction of utrophin through its interaction with the IRES element in the 5' end of the gene and thereby mediates translation. This will be of major interest to the field as this approach to DMD therapy would be applicable to all patients all over the world. Importantly, the data demonstrate that although utrophin A is expressed highly in slow muscle fibres compared to fast ones, the levels of eEF1A2 are similar making this a potential therapeutic pathway for increasing utrophin A levels in all skeletal muscle. Similar levels are found in mdx and wild type mice. They demonstrate that the effects seen are mediated through eEF1A2 as treatment of eEF1A2 null mice do now show increases in utrophin protein.

These are potentially very important results for the therapy of DMD and the quality of the data is very good. However, there are a few aspects that need to be addressed.

We thank the reviewer for carefully examining our manuscript and providing important comments. We were delighted to read the many positive comments from the reviewer on the interest of our work as well as on the quality of our data.

They used a C2C12 muscle cell line assay to screen FDA approved drugs and determined 6 classes of molecules which increase utrophin protein levels. The results show that this shows up to 2 fold increases in utrophin protein. It would be interesting to see the dose response curves. No information is given about how they determined the optimum dose or even if they are presenting the optimum dose. They just say "clinically relevant dose".

We have included, in supplemental figure 1, dose curves of the 11 top drugs. In fact, we treated skeletal muscle cells with three different doses (the dose used in the screen as well as a lower and a higher dose). With these cell lysates, we probed for eEF1A2 and utrophin A protein levels. Interestingly, some of these drugs induced a dose dependent increase of utrophin A and eEF1A2. Each drug dose from the initial drug screen was based on doses used in clinic by patients as listed in the Drug Bank database³. This information was now added to the manuscript (see p.27 lines 514-515)

It would also be essential to demonstrate that this effect is seen DMD patient cell lines to demonstrate that the same mechanism operates in human cells and therefore can be taken forward to the clinic.

This is an excellent point. To address this, we treated human skeletal muscle cells with Betaxolol and Pravastatin for 24 hours and demonstrated that utrophin A and eeF1A2 protein levels are upregulated. These data nicely support our findings *in vitro* and *in vivo* obtained with mouse samples (see supplemental figure 3 and p.18 lines 335-341)

The paper presents foregrip strength data as an assessment of the effect of 2 of these drugs on muscle after injection into mdx mice. The results show a significant positive effect of the drugs. I wonder why they did not use force drop for their assessments of muscle function as this is the standard test used to assess DMD therapies in the mdx mouse worldwide. It is thus difficult to assess their functional rescue in the mdx mouse compared to other therapeutic approaches for DMD.

We thank the reviewer for this important suggestion. We have performed the requested experiments and believe that the new data obtained from the force drop analysis strongly support our *in vivo* force measurements (see figure 4d, p.14 lines 275-279, p.15 280-281 and p.30-31 for methods description).

Does the utrophin protein at the sarcolemma spread away from the NMJ? This could be shown in longitudinal sections of the treated mdx mouse muscle.

As suggested, we performed immunofluorescence experiments by probing for NMJ with Bungarotoxin and with utrophin A on longitudinal TA muscle sections. This clearly demonstrates that utrophin's localization to the NMJ extends into extrasynaptic regions of muscle fibers in Betaxolol- and Pravastatin-treated mdx mouse muscles (See supplemental figure 2 and p. 16, lines 302-304).

Reviewer #3

In the manuscript entitled “Identification of novel therapeutics that target eEF1A2 and upregulate utrophin A translation in dystrophic muscles” Christine Péladeau et al show that overexpression of eEF1A2 in mdx mouse TA muscles results in an increase in IRES-mediated translation of utrophin A and then identify 11 FDA-approved drugs that increase eEF1A2 and utrophin A protein levels in skeletal muscle cell cultures. They then show that treatment of mdx mice with pravastatin and betaxolol improved the dystrophic phenotype and that pravastatin-induced utrophin A upregulation is attenuated in eEF1A2-null mice. The authors claim that these findings demonstrate that eEF1A2-dependent IRES-mediated translation of utrophin A is an important mechanism of regulating utrophin A expression and suggest that pravastatin and betaxolol could be utilized to treat DMD.

Overall the authors should be commended on their discoveries and for preparing a manuscript that is clearly written and easy to follow. The identification of the eEF1A2 link to utrophin A translation appears to be conceptually novel as do the effects of the 11 identified drugs reported in this manuscript.

We truly appreciate the important comments raised by the reviewer regarding our work. We are pleased to read the positive remarks on the clarity of the manuscript and the novelty of our work.

However, there are a few questions related to data interpretation and the level of scientific rigor applied to some studies, as well as concerns related to statistical methods that should be addressed.

1.) The statistical method(s) used in each figure two-tailed student t-tests vs. one-way ANOVAs should be included in the legend

As requested, we have added this information in each legend of the manuscript.

2.) It is encouraging that the effects of prava are observed at clinically relevant exposures; however, can the authors explain the potential discrepancy between the finding for pravastatin treatment reported in the supplementary figure (difference only seen at 200 nM?) vs. 50nM in the primary screen

This is a good point. The difference between drug doses was observed through distinct technical approaches (notably an ELISA assay and western blot analysis). An ELISA is more sensitive and quantitative and may be able to more easily detect significant differences between the different groups at a lower dose. We now discuss this apparent discrepancy in the manuscript (see p.10, lines 199-204)

3.) Figure 5, are the differences between WT and mdx veh mice statistically different? Can the authors address why eEF1A2 levels appear to increase in the bet experiment and decrease in the prava experiment?

We thank the reviewer for pointing this out. As requested, we added the statistical analysis between WT and vehicle-treated mdx mice in the manuscript. The differences seen in figure 5c and d are not significant.

4.) Figure 6, Figure legend appears to be out of order
We have made the necessary changes in the manuscript. We apologize for this oversight.

5.) Figure 7, the levels of eEF1A2 should also be reported
Thank you for this suggestion. We have now added results from western blot analyses of eEF1A2 levels for wasted and WT mice (see figure 7 a, b and quantification)

6.) What were the number of animals treated in each treatment group and how can we be sure that the study was sufficiently powered to detect an effect?

There were 6-8 mice in each animal groups. Our sample size was calculated based on Charan *et al*⁴. At a power of 90% we require a number of 8 mice and at a power of 80% we require a number of 6 mice to obtain a significant difference of $p=0.05$. Thus, our study is sufficiently powered. We have now added this information in material methods (p. 29 lines 549-550).

7.) It is unclear why only pravastatin was validated in the eEF1A2-null mouse. Did the authors test betaxolol in this model? And what were the results?

Thank you for raising this concern. Unfortunately, the eEF1A2-null mice heterozygotes do not breed very well, and we obtain a very small number of eEF1A2-null homozygote progenies. We thus decided to focus on only one drug since this was mostly a control experiment.

8.) Overall there is a lack of mechanistic insight into how these two drugs are mediating this effect. Why is pravastatin different/unique? Other statins are structurally related and demonstrated greater permeability in muscle.

This is a good point. We now further discuss this in the manuscript (see p. 23, lines 421-425).

9.) Statins modify circulating lipids levels, cholesterol disposition, and inflammatory processes. Could any of these parameters have changed in the reported models? Some in vivo characterization should be conducted. Also, statins have been shown to increase autophagy in similar models, the authors should either interrogate this further, or address in more detail in the discussion.

We thank the reviewer for these suggestions. As recommended, we analyzed the effect of our drug treatments on autophagy. We probed for autophagy markers LC3A/B I and II. Our data show that both treatments have no significant effect on autophagy (see supplemental figure 4 and p.23, line 415-421).

References

1. Miura, P., Thompson, J., Chakkalakal, J. V., Holcik, M. & Jasmin, B. J. The Utrophin A 5'-Untranslated Region Confers Internal Ribosome Entry Site-mediated Translational Control during Regeneration of Skeletal Muscle Fibers. *J. Biol. Chem.* **280**, 32997–33005 (2005).
2. Miura, P. *et al.* The utrophin A 5'-UTR drives cap-independent translation exclusively in skeletal muscles of transgenic mice and interacts with eEF1A2. *Hum Mol Genet* **19**, 1211–1220 (2010).
3. Wishart, D. S. *et al.* DrugBank 5.0: a major update to the DrugBank database for 2018. *Nucleic Acids Research* **46**, D1074–D1082 (2018).
4. Charan, J. & Kantharia, N. D. How to calculate sample size in animal studies? *J Pharmacol Pharmacother* **4**, 303–306 (2013).

Reviewers' Comments:

Reviewer #1:

Remarks to the Author:

The authors have satisfactorily addressed my comments in the revised manuscript

Reviewer #2:

Remarks to the Author:

The authors have successfully generated considerable amounts of data to address all of the concerns of the reviewers. The paper is much improved and with the human cell line data is now more relevant for advancing to the clinic.

Reviewer #3:

Remarks to the Author:

The authors have adequately addressed most of the reviewers' concerns and the manuscript is greatly improved. I have no further comments or suggestions.

Response to Reviewers

Reviewer Comments:

Reviewer #1 (Remarks to the Author):

The authors have satisfactorily addressed my comments in the revised manuscript

Reviewer #2 (Remarks to the Author):

The authors have successfully generated considerable amounts of data to address all of the concerns of the reviewers. The paper is much improved and with the human cell line data is now more relevant for advancing to the clinic.

Reviewer #3 (Remarks to the Author):

The authors have adequately addressed most of the reviewers concerns and the manuscript is greatly improved. I have no further comments or suggestions.

Answer to reviewers:

We would like to thank the reviewers for carefully reading our manuscript and providing important insights that increased the quality of our work. We were delighted that all reviewers were unanimously satisfied with our revisions.